# In Vitro Cytotoxic Potential of Selected Jordanian Flora and Their Associated Phytochemical Analysis

**DOI:** 10.3390/plants12081626

**Published:** 2023-04-12

**Authors:** Manal I. Alruwad, Manal M. Sabry, Abdallah M. Gendy, Riham Salah El-Dine, Hala M. El Hefnawy

**Affiliations:** 1Department of Pharmacognosy, Faculty of Pharmacy Cairo University, Cairo 11562, Egypt; 2Department of Pharmacology and Toxicology, Faculty of Pharmacy, October 6 University, Giza 12585, Egypt

**Keywords:** *Senecio leucanthemifolius*, *Clematis cirrhosa*, cytotoxicity, phytochemical estimation

## Abstract

Traditional medicines are a significant source of phytochemicals with potential anticancer effects. Ten Jordanian plants were chosen to be tested for cytotoxicity on human colorectal (HT-29) and breast adenocarcinoma (MCF-7) cell lines. The ethanol extracts were screened for their potential cytotoxic effects using a Sulforhodamine B (SRB) colorimetric assay, using doxorubicin as positive control. Plants extracts exhibiting marked cytotoxic activity were further investigated by qualitative and quantitative phytochemical methods. Total phenolics were quantified using the Folin-Ciocalteu reagent, while flavonoids were quantified using aluminum chloride. The total saponins of the *n*-butanol fraction were estimated using diosgenin as a standard. The total alkaloids and total terpenoids were also evaluated using the gravimetric method. As results, *Senecio leucanthemifolius* (IC_50_: 13.84 μg/mL) and *Clematis cirrhosa* (IC_50_: 13.28 μg/mL) exhibited marked cytotoxic effects on human colorectal adenocarcinoma (HT-29) cell lines. Total phenolics, flavonoids, saponins, alkaloids, and terpenoids found in *Senecio leucanthemifolius* were (91.82, 14.90, 14.27, 101, and 135.4 mg/g of dry extract), respectively. They were revealed to be (68.18, 7.16, 31.25, 73.6, and 180 mg/g of dry extract) in *Clematis cirrhosa*, respectively. *Senecio leucanthemifolius* and *Clematis cirrhosa* have been found to possess cytotoxicity against colorectal (HT-29). In conclusion, the findings of this study offer a new perspective on Jordanian plant extracts anticancer activity research.

## 1. Introduction

Cancer is considered as one of the main causes of death worldwide [1]. In Jordan, breast cancer is the main cause of mortality, contributed 20.8% of all deaths, followed by colorectal cancer (10.9%) and lung cancer (9.1%) [2]. Table 1 illustrates the most prevalent cancer types among Jordanian populations of both sexes and all ages in 2020. Notably, over the past 20 years, both the prevalence and mortality by malignant tumors, in particular breast and colorectal malignancies, have increased among Arabians [3]. There are numerous cancer treatment options, including surgery, radiation therapy, chemotherapy, immunotherapy, cancer vaccines, photodynamic, stem cell transformation, and combinations thereof, all of which have serious side effects. However, these conventional and well-established cancer treatments are constrained by a number of risk concerns, such as the potential for tumor cells to acquire resistance and harmful side effects from chemotherapy such as limited bioavailability, toxicity, no specificity, rapid clearance, and metastasis [4,5,6]. The need for safe and effective anticancer drugs has turned our attention to researching folk medicinal plants in the hope of developing an efficient cytotoxic drug. Many medicinal plants have been shown to have anticancer properties [2]. Various research organizations are targeting medicinal crops for their cytotoxic activity, which can result in the discovery of new anticancer medications. More than half of the FDA-approved medicines used in cancer treatment come from natural products [7]. Some of these include irinotecan, topotecan, and camptothecin from the wood and bark of *Camptotheca acuminata,* etoposide, paclitaxel from *Taxus brevifolia* bark, and the vinca alkaloids vinblastine and vincristinefrom *Catharanthus roseus* in addition to bacterian actinomycin D and mitomycin C as well as bleomycin of marine origin [7,8].

Jordan’s varied geographical and climate conditions result in a wide variety of wild plants [9]. Around 2543 plant species divided into 142 families and 868 genera. There are 485 species from 99 different families that are classified as medicinal plants, and they are dispersed widely around the nation [10]. Despite the availability of modern treatment, traditional medicine is ingrained in Jordanian culture, and herbal therapy has remained popular. According to literature reviews based on the published studies, 27 plant species are regarded as traditional remedies in Jordan and nearby countries for the treatment of various cancers [11].

In this study, ten traditionally used Jordanian plants, collected from the Ajloun highlands (North of Jordan), belonging to four plant families, were selected for this research. Traditional uses for these plants are listed in Table 2. Several previous studies have investigated the medicinal value of these plants, particularly the hepatoprotective effects and free radical scavenging of the ethanol extract of *Onopordum alexandrinum* [12], and the in vitro neuroprotective, antihyperglycemic, and antioxidative effects of *Anchusa undulata* [13,14,15]. The aqueous extract of *Anchusa strigosa* showed antibacterial and antidiabetic activities in diabetic rats [16]. The essential oil of *Glebionis segetum* exhibited antibacterial, antioxidant, in vitro antiproliferative, and acetyl cholinesterase inhibition activities [17]. Antifungal activity, antioxidant activity, and enzyme inhibitory effects of the methanol and hydro methanol extracts of *Clematis cirrhosa* were previously assessed [18,19], as well as the antioxidant capacity, and antimicrobial activity of *Lagoecia cuminoides* [20]. The essential oil of *Eryngium glomeratum* aerial parts and roots showed antibacterial activity [21]. Additionally, *Senecio leucanthemifolius’s* in vitro antibacterial and hypoglycemic activities were previously assessed [22].

The current study aims to assess the in vitro cytotoxic activity of ten selected plants in Jordan on colorectal (HT-29) and breast cancer (MCF-7) cell lines using the sulforhodamine B assay. Additionally, total phenolics, total flavonoids, and total saponin will be colorimetrically quantified; total alkaloids and total terpenoids will also be evaluated using gravimetric methods.

## 2. Results and Discussion

### 2.1. Percentage Yield of the 70% Ethanol Extracts

A high variation in the percentage yield of extracts was observed among the different plants. The results are displayed in Table 3. *Clematis cirrhosa* extract was the highest, with a percentage yield of 23.6%, followed by *Ranunculus asiaticus* 16.8%, while *Lagoecia cuminoides* had the lowest yield, with only 2%.

### 2.2. In Vitro Cytotoxicity

The cytotoxic effects of the 70% ethanol extracts of the plant species under study were evaluated on two cell lines (MCF-7 and HT-29). A dose-dependent response was observed. Applying a gradual decrease to the concentrations of 70% ethanol extracts (100–0.01 μg/mL) to both cell lines showed an increase in the average percentage of survival. Half-maximal inhibitory concentrations (IC_50_ values) are summarized in Table 4. According to the American National Cancer Institute (NCI), optimally crude herbal extracts are considered effective antineoplastic if their IC_50_ < 20 µg/mL or 10 µM after two or three days of incubation [30]. Importantly, significant cytotoxic activity was observed with the 70% ethanol extracts of *S. leucanthemifolius* (IC_50_ of 13.84 μg/mL) and *C. cirrhosa* (IC_50_ of 13.28 μg/mL) extracts against HT-29 cells, while they showed low activity against MCF-7 (IC_50_ value of 56.27 and >100 μg/mL, respectively), compared to doxorubicin (IC_50_ of 0.49 μg/mL and 0.28 μg/mL for HT-29 and MCF-7, respectively). The rest of the tested plant extracts showed no or very low cytotoxic activity against the previously mentioned cell lines. These findings are consistent with earlier reports of cytotoxicity of other *Clematis* spp. Previous research showed that *Clematis apiifolia* methanol extract exhibited good cytotoxicity against mouse lymphocytic leukaemia (L1210), human leukaemia (HL-60), and human ovarian (SK-OV-30) tumor cell lines [31]. Additionally, the extract of *Clematis manshrica* showed anticancer activity on murine sarcoma (S-180), hepatocellular carcinoma (Hepa), and lymphoma (P388) implanted in mice. Several *Senecio* spp. were found to have cytotoxic activity in the literature. The methanol extract of *Senecio gibbosus* subsp. gibbosus demonstrated good cytotoxic activity against human prostate cancer cell lines (LNCaP) and human breast cancer cell lines (MCF-7). The cytotoxic activity of dichloromethane and ethyl acetate fractions was the highest, notably on the human prostate cancer cell lines [32]. The ethyl acetate fraction of *Senecio glaucus* L. aerial parts showed potent cytotoxicity against the human pancreatic cancer cell line (PANC-1) [33]. The bioactive plants’ cytotoxic potential is most likely attributed to the presence of various phytochemicals such as phenolics, flavonoids, saponins, terpenoids, and alkaloids detected by qualitative phytochemical analysis.

### 2.3. Qualitative Phytochemical Screening

The phytochemical screening was performed on the two selected biologically active plants (*S. leucanthemifolius* and *C. cirrhosa*) showing cytotoxic activity against HT-29 cells. Phytochemical screening of their 70% ethanol extracts revealed the presence of phenolics, flavonoids, terpenoids, alkaloids, and saponins. Furthermore, the extracts were subjected to evaluation of these major phytochemicals.

### 2.4. Quantitative Estimation of Some Major Phytochemicals

Based on the preliminary phytochemical screening results, quantitative estimation of *S. leucanthemifolius* and *C. cirrhosa* was carried out using standard methods for key phytochemicals such as phenolics, flavonoids, terpenoids, alkaloids, and saponins. The total phenolic content, expressed as gallic acid, as deduced by the Folin-Ciocalteu test with the pre-established standard calibration curve (Figure 1), was calculated as follows: y = 0.004x + 0.089, R^2^ = 0.994. Where y denotes absorbance, x denotes the corresponding concentration (µg/mL) and R^2^ is the correlation coefficient. The total phenolic content of the 70% ethanol extracts of *S. leucanthemifolius* and *C. cirrhosa* were found to be 91.82 ± 5.68 and 68.18 ± 1.8 mg gallic acid/g of dry extract, respectively. The total flavonoid content, expressed as quercetin, as deduced by the aluminum chloride method with the pre-established standard calibration curve (Figure 1), was calculated as follows: y = 0.003x + 0.139, R^2^ = 0.998. Where y denotes absorbance, x denotes the corresponding concentration (µg/mL) and R^2^ is the correlation coefficient. The total flavonoids content of the 70% ethanol extracts of *S. leucanthemifolius* and *C. cirrhosa* were found to be 14.9 ± 2.82 and 7.16 ± 2.10 mg quercetin/g of dry extract, respectively. The total terpenoid content was 135.4 ± 5.54 and 180 ± 4.18 mg/g of dry extract in *S. leucanthemifolius* extract and *C. cirrhosa*, respectively, while the total alkaloid was found to be 101 ± 4.50 and 73.6 ± 3.85 mg/g of dry extract, respectively. The total saponins content, expressed as diosgenin, as deduced from the pre-established standard calibration curve (Figure 1), was calculated as follows: y = 0.004x + 0.066, R^2^ = 0.99 where y denotes absorbance, x denotes the corresponding concentration (µg/mL), and R^2^ is the correlation coefficient. The total saponins content of the *n*-butanol fraction of the 70% ethanol of the defatted *S. leucanthemifolius* and *C. cirrhosa* were found to be 14.27 ± 2.82 and 31.25 ± 2.1 mg diosgenin/g of dry extract, respectively. It is well recognized that plants are rich sources for different types of bioactive metabolites that play an important role in the therapy. The bioactive plants’ cytotoxic potential of both plants is most likely attributed to the presence of various phytochemicals such as phenolics, flavonoids, saponins, terpenoids, and alkaloids detected by qualitative phytochemical analysis. Among these metabolites are the phenolic compounds which contribute to the chemotherapy by several compounds either in the actual form or after chemical modification [34]. They are powerful antioxidants that have been associated with lower rates of occurrence and mortality in a variety of human diseases [35]. According to epidemiological research, a diet high in flavonoid-rich fruits and vegetables may be linked to a lower risk of cancer [36]. Flavonoids have also been shown to induce cellular cytotoxicity in cancer cells through the induction of apoptosis [37]. Some phytochemicals in *Senecio* and *Clematis* species were reported to possess cytotoxic activity against different malignant cell lines. Correlations between cytotoxic activity and saponin content, which possesses cytotoxic and chemopreventive functions, were also explored in several review publications [38,39,40]. For instance, in *Clematis argentilucida*, two identified saponins (3*β*-*O*-[*β*-D-ribopyranosyl-(1 → 3)-*α*-L-rhamnopyranosyl-(1 → 2)-*α*-L-arabinopyranosyl] hederagenin-11, 13-dien-28-oic acid and 3*β*-*O*-{*β*-D-ribopyranosyl- (1 → 3)-α-L-rhamnopyranosyl-(1 → 2)-[*β*-D-glucopyranosyl-(1 → 4)]-*β*-D-xylopyranosyl} oleanolic acid) showed significant cytotoxic activity against human leukemia (HL-60), hepatocellular carcinoma (Hep-G2), and glioblastoma (U251MG) cells [41,42]. Plant alkaloids are one of the major types of drugs used in cancer chemotherapy [43]. On the other hand, pyrrolizidine alkaloids are characteristic metabolites present in most of *Senecio* species [44,45]. They exhibited a genotoxic activity [46], a significant cytotoxic potency against many carcinoma cells. Jacaranone, a pyrrolizidine alkaloid isolated from *Senecio ambigus*, was found to be cytotoxic to both prostate carcinoma (LNCaP) and renal adenocarcinoma (ACHN) cell lines [45]. Previous research has demonstrated that *S. stabianus n*-hexane extract is efficacious against melanotic melanoma (C32), hormone-dependent prostate cancer (LNCaP), human breast adenocarcinoma (MCF-7), and renal adenocarcinoma (ACHN0) cancer cell lines regarding the presence of the terpenes linalool and caryophyllene [44]. Likewise, the alkaloids reported in *Clematis* spp. demonstrated extensive cytotoxicity against many cancer cell lines. The alkaloids extracted from an ethanol extract of *Clematis terniflora* aerial parts showed significant cytotoxicity against the human esophageal carcinoma cell line (ECA-109) [47]. Moreover, *Clematis argentilucida* triterpenes were found to be effective against hepatic carcinoma (Hep-G2), glioma (U251MG), and leukemic cells (HL-60) [48].

## 3. Materials and Methods

### 3.1. Plant Material

The plants were harvested from the Ajloun highlands, which consists of the Mediterranean-like hills ranging from 600–1100 m above sea level in Jordan during their flowering season (April 2021). Taxonomic identification of the plant material was confirmed by Mr. Sameh Khatatbeh, ecological researcher and specialist in flora research at the Royal Society for the Conservation of Nature (RSCN) (Amman, Jordan). A specimen of each plant was kept at the Department of Pharmacognosy, Faculty of Pharmacy, Cairo University. A list of the voucher numbers of the plants is provided in Table 2.

### 3.2. Preparation of the Extracts

Each plant’s aerial parts were gathered, air-dried in the shade, ground into a fine powder (mesh size: 0.2–0.636 mm), and kept in firmL y closed glass bottles until use. The powders were macerated in 70% ethanol in three separate extractions with constant shaking. The obtained extracts were evaporated under reduced pressure using a rotary evaporator (Stuart, Staffordshire, UK) till complete dryness and the extract yields were calculated. Table 3 displays the extraction yields calculated as dry weight percentage.

### 3.3. Chemicals and Reagents

Ethanol, sulfuric acid, acetic acid, ammonium hydroxide, methanol, sodium carbonate, sodium nitrite, aluminum nitrate, aluminum chloride, ammonia, sodium hydroxide, sodium chloride, *n*-butanol, diethyl ether, petroleum ether, *p*-anisaldehyde, hydrochloric acid, potassium bismuth iodide, ethyl acetate, etc. were purchased from Biochem, Egypt. Advanced Dulbecco’s modified Eagle’s medium (DMEM), bovine serum, trypsin-EDTA, L-glutamine, fetal bovine serum, penicillin, streptomycin, tris-(hydroxymethyl) aminomethane (TRIS), trichloroacetic acid (TCA), dimethylsulfoxide (DMSO), sulforhodamine B (SRB), gallic acid, Folin and Ciocalteu′s phenol reagent, quercetin, diosgenin, were supplied from Sigma Aldrich (Steinheim, Germany), doxorubicin hydrochloride (Ebewe Pharma, Unterach, Austria).

### 3.4. Cytotoxicity Screening

#### 3.4.1. Cell Culture

Two human cancer cell lines, HT-29 and MCF-7, were used for cytotoxicity screening of the ten selected Jordanian medicinal plant extracts. Both cell lines were provided from Nawah Scientific Inc. (Mokatam, Cairo, Egypt). Cells were retained in DMEM media, supplemented with 100 mg/mL of streptomycin, 100 units/mL of penicillin, and 10% of heat-inactivated fetal bovine serum in a humidified 5% (*v*/*v*) CO_2_ atmosphere at 37 °C. Doxorubicin was used as a positive control.

#### 3.4.2. In Vitro Cytotoxicity Assay

Cell viability was estimated by the SRB assay, adopting the method of El Haddad et al. 2022 [49]. Aliquots of 100 μL cell suspension (5 × 10^3^ cells) were put in 96-well plates and incubated in complete media for 24 h. A second aliquot of 100 μL of medium with different doses of the extracts was applied to the cells. Cells were fixed by replacing the medium with 150 μL of 10% TCA and incubating at 4 °C for 1 h after 72 h of plant extract exposure. The TCA solution was withdrawn and distilled water was used to wash the cells five times. Aliquots of 70 μL SRB solution (0.4% *w*/*v*) were added and incubated for 10 min in a dark place at room temperature. Plates were washed 3 times with 1% acetic acid before being let to air dry overnight. Finally, 150 μL of tris(hydroxymethyl) aminomethane (TRIS) (10 mM) were added to dissolve protein-bound SRB stain; the optical density (OD) of both treated and untreated cells was read at 540 nm. The percentage growth inhibition of cells exposed to treatments was calculated as follows.
%Inhibition = ([OD of control − OD of sample]/OD of control) × 100

#### 3.4.3. Determination of IC_50_ Values

The toxicity percent of each treated well was calculated, and since the experiments were performed in triplicate, the average toxicity percent was also calculated for every treatment concentration. Then, a curve of cytotoxicity percentage against the logarithm of concentration was drawn, and the IC_50_ (concentration at which there was 50% cell death compared to negative control) value was calculated from the equation of the logarithmic line.

### 3.5. Data Analysis

All the tests were conducted in triplicates, and the data were analyzed using one-way analysis of variance (ANOVA). A Student’s *t*-test was then performed to determine the statistical significance of the difference between control and plant extract treated groups, with a level of significance (*p* < 0.05).

### 3.6. Qualitative Phytochemical Analysis of Active Extracts

A qualitative phytochemical screening of *S. leucanthemifolius* and *C. cirrhosa* extracts was performed using standard methods described by Evans 2009 [50]. The powder of each plant sample was extracted with water and 70% ethanol at room temperature with constant shaking. Saponins were tested in the aqueous extract, while phenolics, flavonoids, alkaloids, terpenoids, and saponins were tested in the 70% ethanol alcoholic extract.

### 3.7. Quantitative Phytochemical Analysis

#### 3.7.1. Determination of the Total Phenolics

The total phenolic contents of *S. leucanthemifolius* and *C. cirrhosa* extracts were spectrophotometrically determined using the Folin–Ciocalteu method as described by Attard 2013 [51]. Samples preparation was performed at a concentration of 5 mg/mL in methanol. In brief, an addition of 10 μL of sample/standard to 100 μL of Folin-Ciocalteu reagent (Diluted 1:10) (*v*/*v*) was applied in a 96-well microplate. Subsequently, the mixture was incubated for 20 min in the dark at room temperature (25 °C) with 80 μL of 1M Na_2_CO_3_. After the incubation time ends, microplate reader FluoStar Omega (Ortenberg, Germany) was used to measure the developing complex blue color at 630 nm. Data are displayed as means ± SD. Eight serial dilutions of the 1 mg/mL gallic acid stock solution in methanol were prepared in the concentrations of 0.025, 0.05, 0.10, 0.20, 0.40, 0.60, 0.80, and 1 mg/mL.

#### 3.7.2. Determination of the Total Flavonoids

The total flavonoid contents of *S. leucanthemifolius* and *C. cirrhosa* extracts were spectrophotometrically determined using the aluminum chloride method as described by Kiranmai et al. 2011 [52]. Ten serial dilutions of the 1 mg/mL quercetin stock solution in DMSO were prepared in the concentrations of 0.001, 0.002, 0.004, 0.008, 0.025, 0.050, 0.10, 0.20, 0.40, and 0.80 mg/mL. Samples were prepared at a concentration of 5 mg/mL in methanol. In brief, a total of 15 μL of sample/standard was placed in a 96-well microplate. After that, 175 μL of methanol and 30 μL of 1.25% AlCl_3_ were added, followed by addition of 30 μL of 0.125 M C_2_H_3_NaO_2_ and incubation for 5 min. A microplate reader was used to measure the ensuing yellow color at 420 nm at the end of the incubation period. Data are presented as means ± SD.

#### 3.7.3. Determination of the Total Terpenoids

The total terpenoids were determined gravimetrically. About ten g of *S. leucanthemifolius* and *C. cirrhosa* powder were macerated in 70% alcohol for 24 h. The extract was subjected to filtration then extraction with petroleum ether. The petroleum ether extracts were separately evaporated to dryness into pre-weighed glass. The total terpenoids yield (%) was calculated according to Indumathi et al. 2014 [53].

#### 3.7.4. Determination of Total Alkaloids

The total alkaloids were determined gravimetrically. In a 250 mL beaker, 5 g of *S. leucanthemifolius* and *C. cirrhosa* powder were weighed, followed by the addition of 200 mL of 10% ethanolic acetic acid, capped, and left to stand for four hours. After filtration of the samples, the extracts were concentrated in a water bath to about the quarter of their volume. In order to fully precipitate the extract, concentrated ammonium hydroxide was added dropwise. After allowing the resulting suspension to settle, the precipitated material was washed with diluteNH**_4_**OH, then dried and weighed according to Harborne 1998 [54].

#### 3.7.5. Determination of Total Saponins

The concentrated 70% ethanol extract was suspended in distilled water, defatted twice with diethyl ether (10 mL each) in a separating funnel, then the ethereal layer was rejected. The remaining aqueous layer was extracted with *n*-butanol saturated with water (2 × 10 mL). The combined *n*-butanol fraction was washed with 1 mL of 5% aqueous NaCl solution, then concentrated to drying. A stock solution of standard Diosgenin in ethyl acetate (1 mg /mL) was prepared. A weighed amount of the dried *n*-butanol fraction of the 70% ethanol extracts of the defatted *S. leucanthemifolius* and *C. cirrhosa* were dissolved in methanol to yield a 0.20 mg/mL solution.

The total saponin contents of *S. leucanthemifolius* and *C. cirrhosa* in the *n*-butanol fraction were determined spectrophotometrically, using diosgenin as a standard, as described by Saxena et al., 2013 [55]. The diosgenin level was determined in the tested sample by measuring the absorbance (at λ = 430 nm) of the color produced as a result of the reaction of steroidal saponins with *p*-anisaldehyde/sulfuric acid in an ethyl acetate medium.

For the establishment of the calibration curve, serial dilutions of diosgenin (0.01–0.08 mg/mL) were prepared in ethyl acetate. An aliquot (one mL) of each dilution was added to two mL of ethyl acetate in a test tube, treated with one mL of each of reagent A (*p*-anisaldehyde (99%)/ethyl acetate 0.5:99.5 *v*/*v* ethyl acetate) and B (concentrated sulfuric acid/ethyl acetate 50:50 *v*/*v*) and stirred. The reaction mixture was heated at 60 °C for 10 min, then allowed to cool for another 10 min at ambient temperature. The absorbance of the greenish-violet color produced was measured at 430 nm against a blank prepared with pure ethyl acetate using a double-beam spectrophotometer UV-1650 PC (Shimadzu, Kyoto, Japan). Each determination was carried out in triplicate, and the calibration curve was constructed by plotting the average of the observed absorbance versus concentrations.

## 4. Conclusions

The findings of this study offer a new perspective on Jordanian plant extracts anticancer activity research. Some Jordanian plant extracts demonstrated a significant anti-proliferative effect with lower IC_50_ values against human colorectal (HT-29) carcinoma cell lines, for example, 70% ethanol extracts from *Senecio leucanthemifolius* with an IC_50_ value of 13.84 μg/mL, and the ethanol extract from *Clematis cirrhosa* with an IC_50_ value of 13.28 μg/mL. These values are below the range of 30 µg/mL that the national cancer institute considers interesting for raw plant extracts. These cytotoxic results revealed that both extracts can trigger apoptosis in culture. Quantitative estimation of total phenolics, flavonoids, saponins, alkaloids, and terpenoids found in *Senecio leucanthemifolius* were (91.82, 14.90, 14.27, 101, and 135.4 mg/g of dry extract), respectively. They were revealed to be (68.18, 7.16, 31.25, 73.6, and 180 mg/g of dry extract) in *Clematis cirrhosa*, respectively. In recent years, several phytochemicals extracted from therapeutic plants with potential anticancer properties have been discovered. In order to increase the number of cytotoxic compounds for potential medication development, rigorous screening must be conducted. Moreover, the use of plant extracts and/or their bioactive compounds alone or in combination with traditional chemotherapy may be anticipated in the future as promising adjuvant drugs in cancer therapy. However, more research is required to isolate bioactive compounds from plant extracts and assess their unknown effects as well as their synergistic effects, in in vitro and in vivo animal models. Further in vivo studies are required to fully understand the impact of the reported results.

## Figures and Tables

**Figure 1 plants-12-01626-f001:**
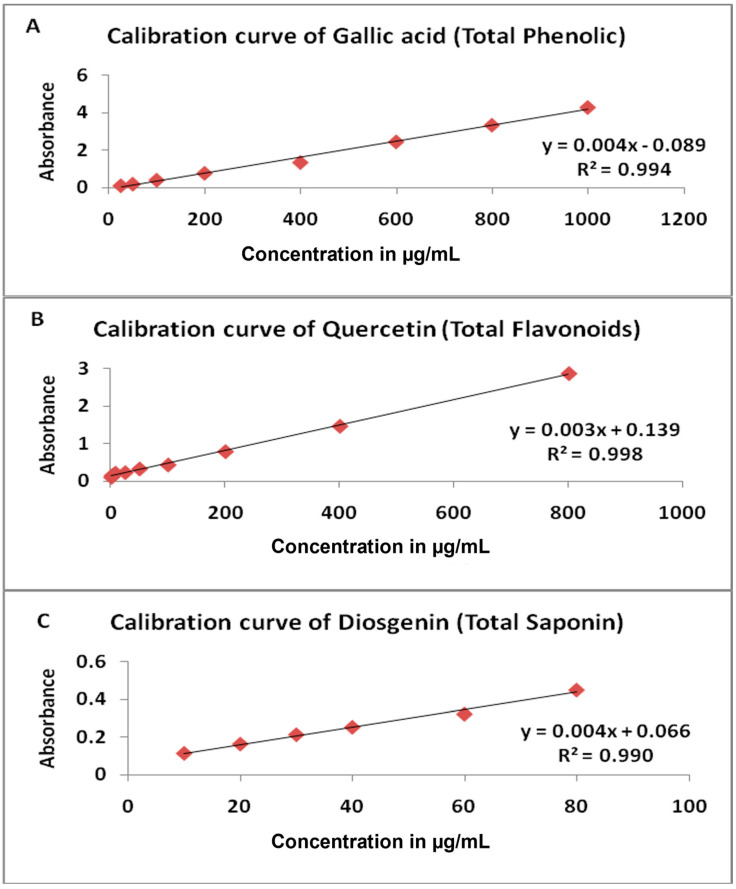
Standard calibration curve (**A**) Gallic acid, (**B**) Quercetin, (**C**) Diosgenin.

**Table 1 plants-12-01626-t001:** The most common cancers in Jordan, both sexes, all ages, in 2020.

Rank	Site	Number of Cases	Percent
1	Breast	2403	20.8%
2	Colorectal	1260	10.9%
3	Lung	1047	9.1%
4	Urinary Bladder	572	4.9%
5	Leukemia	569	4.9%

**Table 2 plants-12-01626-t002:** A list of the plant species evaluated in this study, names and traditional uses.

Plant Scientific Name	Family	Common Names (English)	Local Name (Arabic)	Traditional Uses	Specimen No.
*Glebionis segetum* L. Fourr.	Asteraceae	Corn Marigold	(Albusbas)	Treatment for fever, nocturnal sweats, spasms, burns, and ulcers. Diuretic and stomachic [23]	17-06-2021 Ι
*Onopordum alexandrinum* Boiss.	Asteraceae	Cotton Thistle	(Atour)	Hepatoprotective [24]	19-06-2021 Ι
*Phagnalon rupestre* L. DC.	Asteraceae	African Fleabane	(Qadaha)	Abdominal pain Joints pain [23,25]	16-06-2021 Ι
*Senecio leucanthemifolius* Poir.	Asteraceae	Eastern Groundsel	(Shykha)	Cancer treatment [26]	21-06-2021 Ι
*Clematis cirrhosa* L.	Ranunculaceae	Evergreen Virgin´s-Bower	(Hublmiskiun)	Skin diseases [19]	20-06-2021 Ι
*Ranunculus asiaticus* L.	Ranunculaceae	Turban Buttercup	(Hawdhan)	Pertussis [27]	18-06-2021 Ι
*Anchusa strigosa* Banks & Sol.	Boraginaceae	Prickly Alkanet	(Humhum)	Astringent for burns and wounds, anti-ulcer, cough, and rheumatic inflammation [23]	14-06-2021 A
*Anchusa undulata* L.	Boraginaceae	Common Alkanet	(Humhum)	Antidiabetic [15]	14-06-2021 B
*Eryngium glomeratum* Lam.	Apiaceae	Clustered eryngo	(Euidalqizm)	Scorpion and snakes bite. Diuretic, renal stones [23,28]	22-06-2021 Ι
*Lagoecia cuminoides* L.	Apiaceae	False Cumin	(Amjiris)	Improving the digestion Bilestone repellent [29]	15-06-2021 Ι

**Table 3 plants-12-01626-t003:** Percentage Yield of the screened ethanol extracts.

Name of the Plant	Yield (%)
*Glebionis segetum* L. Fourr.	7.7%
*Onopordum alexandrinum* Boiss.	9.2%
*Phagnalon rupestre* L. DC.	6.2%
*Senecio leucanthemifolius* Poir.	10.4%
*Clematis cirrhosa* L.	23.6%
*Ranunculus asiaticus* L.	16.8%
*Anchusa strigosa* Banks & Sol.	13.6%
*Anchusa undulata* L.	8.4%
*Eryngium glomeratum* Lam.	6.4%
*Lagoecia cuminoides* L.	2%

**Table 4 plants-12-01626-t004:** Cytotoxic activity (IC_50_) of plant extracts under investigation towards HT-29 and MCF-7 cell lines.

Name of the Plant	IC_50_ Values of Cell Lines (μg/mL)	
	HT-29	MCF-7
*Glebionis segetum* L. Fourr.	203.90 μg/mL mL ± 17.11	>100 μg/mL
*Onopordum alexandrinum* Boiss.	120.24 μg/mL ± 3.55	234.79 μg/mL ± 10.98
*Phagnalon rupestre* L. DC.	>100 μg/mL	>100 μg/mL
*Senecio leucanthemifolius* Poir.	13.84 μg/mL ± 0.14	56.27 μg/mL ± 1.02
*Clematis cirrhosa* L.	13.28 μg/mL ± 0.69	>100 μg/mL
*Ranunculus asiaticus* L.	175.94 μg/mL ± 2.25	656.47 μg/mL ± 20.77
*Anchusa strigosa* Banks & Sol.	188.84 μg/mL ± 12.91	191.48 μg/mL ± 5.67
*Anchusa undulata* L.	534.59 μg/mL ± 15.21	>100 μg/mL
*Eryngium glomeratum* Lam.	68.51 μg/mL ± 3.46	173.45 μg/mL ± 5.31
*Lagoecia cuminoides* L.	617.53 μg/mL ± 7.50	>100 μg/mL
Doxorubicin	0.49 μg/mL ± 0.05	0.28 μg/mL ± 0.02

Results are the average and standard deviation of three replicates. Doxorubicin: positive control.

## Data Availability

No new data were created or analyzed in this study. Data sharing is not applicable to this article.

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
