# Peer review of "In Vitro Cytotoxic Potential of Selected Jordanian Flora and Their Associated Phytochemical Analysis"

_plants, 2023, doi:10.3390/plants12081626_

Round 1
Reviewer 1 Report
The article "In Vitro Cytotoxic Potential of Selected Jordanian Flora and Their Associated Phytochemical Analysis" offers a new perspective on the anticancer potential, and not only of 10 Jordanian plants.
Total phenols, flavonoids, saponins, alkaloids and terpenoids have also been reported for these plants.
The best antitumor effects are marked both in the Abstract and in the conclusions, with the necessary discussions in the content of the work.
The work is interesting and carefully written and deserves to be published in its current form.

Author Response
Please accept our revised manuscript, we would like to thank the reviewer for his/her constructive comments that have helped us significantly to improve the manuscript.

Reviewer 2 Report
Comments:
1. Page 1, line 19-20, in the “Abstract”, “IC50” should be amended “IC50”.
|
2. Page 2, in the “Table 2”, “(Ali‐Shtayeh and Abu Ghdeib, 1999)” and “(Qasem, 2015; Wang et al., 2012)” should be amended “[40]” and “[41][42]”. |
3. Page 4, line 103, “Senecio gibbosus subsp. gibbosus” should be amended “Senecio gibbosus subsp. gibbosus”.
4. Page 5, line 154, “(Shibata, 2001; Kerwin, 154 2004; Bachran et al., 2008).” should be amended “[28][29][30].”.
5. Page 5, line 163, “(Tundis et al., 2009).” should be amended “ [33].”.
6. Page 5, line 169, in the “Figure 1”, There should be a scale and a dose indication on the diagram.
7. Page 5, line 173, “(A) Gallic,” should be amended “(A) Gallic acid,”.
8. Page 7, line 210, “(5x103 cells)” should be amended “(5x103 cells)”.
9. Page 10, line 398, “J Nat Prod 2008, 1, 10–26.” should be amended “J. Nat. Prod. 2008, 1, 10–26.”.
10. Page 11, line 403, “Plants , 11, 1599 2022, 1599.” should be amended “Plants (Basel) 2022, 11, 1599.”.
11. Page 11, line 416, “Int J ChemTech Res 2014, 6, 4264–4267.” should be amended “Int. J. ChemTech. Res. 2014, 6, 4264–4267.”.
12. Page 11, line 419, “Int J Seed Spices 2013, 3, 26–30.” should be amended “Int. J. Seed Spices 2013, 3, 26–30.”.
Author Response
Please accept our revised manuscript, we would like to thank the reviewers for their constructive comments that have helped us significantly to improve the manuscript. Provided below a list answering all raised comments and queries.
Response: All concerns were adjusted; changes were tracked using the “Track Changes” function in MS Word.
- Page 1, line 19-20, in the “Abstract”, “IC50” should be amended “IC50”.
Response: It was corrected
- Page 2, in the “Table 2”, “(Ali‐Shtayeh and Abu Ghdeib, 1999)” and “(Qasem, 2015; Wang et al., 2012)” should be amended “[40]” and “[41][42]”.
Response: It was corrected
- Page 4, line 103, “Senecio gibbosusgibbosus” should be amended “Senecio gibbosus subsp. gibbosus”.
Response: It was corrected
- Page 5, line 154, “(Shibata, 2001; Kerwin, 154 2004; Bachran et al., 2008).” should be amended “[28][29][30].”.
Response: It was corrected
- Page 5, line 163, “(Tundis et al., 2009).” should be amended “[33].”.
Response: It was corrected
- Page 5, line 169, in the “Figure 1”, There should be a scale and a doseindication on the diagram.
Response: It was corrected
- Page 5, line 173, “(A) Gallic,” should be amended “(A) Gallic acid,”.
Response: It was corrected
- Page 7, line 210, “(5x103 cells)” should be amended “(5x103cells)”.
Response: It was corrected
- Page 10, line 398, “J Nat Prod 2008, 1, 10–26.” should be amended “ Nat. Prod.2008, 1, 10–26.”.
Response: It was corrected
- Page 11, line 403, “Plants , 11, 1599 2022, 1599.” should be amended “Plants (Basel) 2022,11, 1599.”.
Response: It was corrected
- Page 11, line 416, “Int J ChemTech Res 2014, 6, 4264–4267.” should be amended “ J. ChemTech. Res.2014, 6, 4264–4267.”.
Response: It was corrected
- Page 11, line 419, “Int J Seed Spices 2013, 3, 26–30.” should be amended “ J. Seed Spices 2013, 3, 26–30.”.
Response: It was corrected

Reviewer 3 Report
Please, refer to PDF file.

Author Response
Please accept our revised manuscript, we would like to thank the reviewers for their constructive comments that have helped us significantly to improve the manuscript. Provided below a list answering all raised comments and queries.
Response: All concerns were adjusted; changes were tracked using the “Track
Changes” function in MS Word.
The manuscript describes the screening of ethanol extracts from ten Jordanian plant species and their in vitro cytotoxicity on human colorectal (HT-29) and breast adenocarcinoma (MCF-7) cell lines. Two extracts with cytotoxic potential using the criteria established by American National Cancer Institute were submitted to phytochemical investigation of the main classes of metabolites.
- The methodology used in bioassays and phytochemical screening are classic and are well described in most stages. Authors need to review the procedures in section 3.6 (Qualitative phytochemical analysis of active extracts).
Response: The methodology section was revised carefully, and all issues were clarified.
- In this section, aqueous and 70% ethanolic extracts were obtained and used in phytochemical screening. However, terpenoids and alkaloids were quantified from powdered plant material. Butanol fractions were originated from aqueous or ethanolic extracts?
Response: This section was revised carefully, and all issues were clarified.
- There are many words in the text that need a separator (space) between them.
Response: the text was revised carefully, and all words were corrected.
- Table 1 does not differentiate genders and should be revised.
Response: The table was revised and corrected to both sexes, all ages.
- The text needs revision in the binomial nomenclature of some species (Clematis cirrhosa L., Ranunculus asiaticus L., Anchusa undulata L. and Lagoecia cuminoides L.).
Response: The text was revised, and the binomial nomenclatures were corrected.
- Units require standardization and all figures can be deleted without compromising the text (linear regression equations and R2 values were cited in the methodology sections).
Response: the text was revised carefully, and all units were corrected.
- Add a comma after “sodium hydroxide” (P6/L193) and use lower case in reagents (P6/L196-197; P7/L198-199).
Response: the text was revised carefully, and all words were corrected.
- Latin expressions should be used as in vitro, in vivo. Change bold text (page 7/lines 220-221) to regular font.
Response: the text was revised carefully, and all words were corrected.
- References also needs revision, especially with regard to standardization in titles (lower case) and using italics for species names in all titles.
Response: All references were revised and corrected as suggested by the reviewer.
- Thus, the manuscript makes a positive contribution as a preliminary step for more advanced studies of the bioactive species and is suitable for Structural and Functional Analysis of Extracts in Plants section
Round 2
Reviewer 2 Report
Comments:
1. Page 10, line 401, in the ref . 26, “Structure-Dependent Cytotoxic effects of eremophilanolide 401 sesquiterpenes. Nat. Prod. Commun. 2017, 12, 1934578X1701200505.” should be amended “Structure-dependent cytotoxic effects of eremophilanolide sesquiterpenes. Nat. Prod. Commun. 2017, 12, 663‒665.”.
2. Page 11, line 403, in the ref. 27, “Plants (Basel), 11, 1599 2022, 1599.” should be amended “Plants (Basel) 2022, 11, 1599.”.
Author Response
Response: All concerns were adjusted; changes were tracked using the “Track
Changes” function in MS Word.
Reviewer comments:
Reviewer #2
- Page 10, line 401, in the ref . 26, “Structure-Dependent Cytotoxic effects of eremophilanolide 401 sesquiterpenes. Prod. Commun. 2017, 12, 1934578X1701200505.” should be amended “Structure-dependent cytotoxic effects of eremophilanolide sesquiterpenes. Nat. Prod. Commun. 2017, 12, 663‒665.”.
Response: It was corrected
- Page 11, line 403, in the ref. 27, “Plants(Basel), 11, 1599 2022, 1599.” should be amended “Plants (Basel) 2022, 11, 1599.”.
Response: It was corrected
